# A Novel Indole Derivative with Superior Photophysical Performance for Fluorescent Probe, pH-Sensing, and Logic Gates

**DOI:** 10.3390/ijms24021711

**Published:** 2023-01-15

**Authors:** Hai-Ling Liu, Kan Zhan, Kai-Liang Zhong, Xing-Liang Chen, Xing-Hua Xia

**Affiliations:** 1School of Chemistry and Chemical Engineering, Shaoxing University, Shaoxing 312000, China; 2State Key Laboratory of Analytical Chemistry for Life Science, School of Chemistry and Chemical Engineering, Nanjing University, Nanjing 210023, China; 3College of Biotechnology and Bioengineering, Zhejiang University of Technology, Hangzhou 310014, China

**Keywords:** indole, fluorescent, photophysical, pH-sensor, logic gates

## Abstract

An indole-related molecules have been considered as the potential fluorescent probes for biological and electrochemical sensing. However, most of the indole probes have been usually used in a single detection mode. Indolium probes that enable accurate detection in complex environments are rarely reported. Here, four novel indole derivatives including the phenyl group substituted with different functional moieties were designed on the basis of the donor-π-acceptor (D-π-A) concept. These derivatives exhibit positive solvatochromism owing to their varied molecular conformations upon contacting to various solvents and the different HOMO-LUMO gaps caused by the difference in electronic push-pull capability of the substituents. Their solid-state fluorescence emissions and multiple chromisms are observed due to the inherent twisted geometries and aggregation modes. In addition, these derivatives show dramatic color and fluorescence responses due to the protonation of the nitrogen and oxygen containing groups, and thus novel colorimetric pH sensors, fluorescent papers and logic gates have been designed.

## 1. Introduction

Fluorescence probe has been considered as a kind of molecular measurement device based on spectrochemistry, optical waveguide and measurement technology, which could selectively and continuously convert the chemical information of analytic targets into fluorescence signals. When the fluorescence probes are stimulated by the surroundings, their fluorescence emission changes allowing scientists to learn about the characteristics of the surroundings or the specific information that existed in the surroundings [1]. In recent years, fluorescent probes have been widely used in such various fields as material, environmental, and life and information science, etc. [2,3,4,5] because they have multiple benefits such as high sensitivity, good selectivity, fast and convenient use, low cost, no pretreatment, no influence of external electromagnetic field, and long-distance luminescence. [6,7,8,9]. As a class of powerful candidates for excellent optical probes, organic conjugated small molecules with electron donor-π-acceptor (D-π-A) architectures have been the focus of attention and research [10,11,12,13,14]. Among them, indole, triphenyl boride [15,16,17], rhodamine [18,19], and coumarin [20,21,22] have been reported frequently.

Indole and its derivatives have strong fluorescence emission in solution, and thus can be used as good fluorescent probes in the biological or electrochemistry detecting. For example, a novel indole-based fluorescent probe containing amino group and hemicyanine moiety has been explored for the recognition and detection of hypochlorite (OCl^−^) in the living systems as well as the bioimaging in live cells [23]. A novel chemodosimeter based on naphthalimide-indole ion conjugate molecule has been synthesized for selectively sensing cyanide in the presence of other anions in an aqueous medium [24]. Further, a ratiometric fluorescent probe with a 3-formyl-BODIPY as fluorophore group and indole salt as a recognition group has been prepared for the specific detection of the cyanide ion [25]. In addition, indole and its derivatives are also of the special response to protons due to the existed N atoms in their structures. For this reason, some acidic pH probes with related structures have stood out. For instance, the styrylcyanine-based pH probes have been developed via ethylene bridging of the non-N-substituted indole derivatives and 4-(diphenylamino) benzaldehyde. Additionally, their remarkable pH-dependent behavior in vis-NIR fluorescence emission is especially suitable for in vivo imaging because they could effectively minimize photodamage and avoid the influence of cell auto-fluorescence [26].

Unfortunately, most of indole probes usually only follow a single detection mechanism for the detected object by now, and few publications concern the multilevel signal response especially responses in various possible complex detection environments. With the development of society, the versatility of indole molecules needs to be further enhanced in order to meet the actual detection requirements. Meanwhile, the functional molecules based on indole with aggregation-induced emission (AIE) property are rarely studied, and there is little discussion about how to regulate their AIE features. In order to realize the transition from molecule design to material application, it is of great theoretical and practical importance for studying the properties in their aggregated state such as solid powders and crystals for these functional molecules.

In this work, we designed four novel fluorescent molecules, 5-(4-ethoxyphenyl)-2,3,3-trimethyl-3H-indole (**PI**), *N*,*N*-dimethyl-4-(2,3,3-trimethyl-3H-indol-5-yl)aniline (**NI**), *N*,*N*-diphenyl-4-(2,3,3-trimethyl-3H-indol-5-yl)aniline (**TI**), and 2,3,3-trimethyl-5-(4-(trifluoromethyl)phenyl)-3H-indole (**FI**), by attaching different phenyl-based derived structural units to the indole moiety. The effects of difference in electropositivity for the four functional groups (p-ethoxyphenyl, 4- (*N*,*N*- dimethylamine) phenyl, 4- triphenylamine and trifluoromethylphenyl) in **PI**, **NI**, **TI** and **FI**, respectively. Additionally, the solvatochromic behaviors, aggregation-induced emission features, luminescence performance and multiple chromisms in solid and crystalline state were discussed. The theoretical calculation result of these molecules with a similar framework of D-π-A structure in crystalline state was analyzed, and the relationship between electronic structures and molecular energy levels was studied. In addition, for these as-prepared new functional molecules, their potential application for pH-sensor in solution and acidic atmosphere, fluorescent paper, and logic gates were investigated.

## 2. Results and Discussion

### 2.1. Synthesis

In this work, four kinds of new fluorescent compounds consisting of phenyl-based derivatives and indole (**PI**, **NI**, **TI** and **FI**) were designed and synthesized by a simple synthetic route with a high yield (Figure 1). All the intermediates and final products were purified carefully and characterized fully by ^1^H NMR, ^13^C NMR, HRMS and IR, which confirm their expected molecular structures (see the Appendix A). These compounds are soluble in common organic solvents such as THF and DMF, but insoluble in water.

### 2.2. Solvatochromism

Fluorescence emission spectra of **PI**, **NI**, **TI,** and **FI** in different solvents are shown in Figure 1, which shows that the four compounds possess different degrees of solvation effect. In non-polar hexane, the main emission peaks of **PI**, **NI**, **TI,** and **FI** appear at 499 nm, 438 nm, 390 nm, and 399 nm, respectively. Beyond that, one or two shoulder peaks also emerge, which indicated two or three close-lying excited vibration states with small energy gap. The variation of the fluorescence spectra for the change of solvent polarity probably related to the potential energy surfaces of the molecule in diverse settings, these results are similar to the previous reported references [27]. In comparison with **PI** molecule, **NI**, **TI,** and **FI** molecules all show a blue shift in fluorescent emission. For **FI**, this was because the electron-withdrawing effect of trifluoromethyl phenyl group reduces the conjugation degree of electron cloud of whole **FI** molecule [28]. As for **NI** and **TI**, although *N*,*N*-dimethylamine phenyl group in **NI** and triphenylamine group in **TI** could also provide the lone pair electrons and aromatic rings to enlarge conjugate surface, their large steric hindrance makes whole molecular structure distorted, resulting in a decrease of conjugation degree of electron cloud and thus a blue shift of fluorescence emission [29,30,31,32]. With increasing solvent polarity, red-shifted emissions with varying degrees are observed in mid-polar solvents (EA, THF and DCM) and polar solvents (DMF and acetonitrile). For instance, the fluorescence peaks of **PI**, **NI**, **TI,** and **FI** are red shifted by 50 nm, 29 nm, 55 nm, and 57 nm, respectively, suggesting the presence of positive solvatochromism. In this regard, the differences in intramolecular electron push-pull capabilities and molecular polarity should be the main responsibility. As polarity of the solvents increases, the molecular conformation changes gradually from planar to tortile, and their electrons are also gradually separated from donors and acceptors, leading to twisted intramolecular charge transfer (TICT) [33].

### 2.3. Solid State Luminescence Property and Multiple Chromism (MC) Effect

The fluorescence peaks of **PI**, **NI**, **TI,** and **FI** in the precipitation solid state could be found at 583 nm, 471 nm, 472 nm, and 523 nm, respectively under the ultraviolet lamp (shown in Figure 2). Since **PI** possesses the strongest electrons push-pull ability at solid state, its fluorescence emission appears in the longest wavelength region. By comparison, the fluorescence peaks of **NI**, **TI,** and **FI** molecules undergo blue shifts in varying degrees due to the difference of intramolecular electron push-pull effect.

After dissolving **PI**, **NI**, **TI,** and **FI** samples in tetrahydrofuran/ethanol (*v*/*v* = 1:1) mixed solution and then slowly volatilizing the solvent, the obtained crystal samples of **PI-C**, **NI-C**, **TI-C,** and **FI-C** show a big difference from their respective precipitated samples in PL spectra and emission color (Figure 3). The solid fluorescence emission peaks of the series of synthesized compounds range from 451 nm to 612 nm, covering a wide range of color (Figure 4), which could meet the requirements of fluorescent materials for different fluorescence emission wavelengths to a large extent [34].

The wavelength of fluorescence peaks for crystallization samples of **PI**, **NI,** and **FI** has a rather clear blue shift in different degrees, respectively compared to their precipitation samples, while the fluorescence emission peak of **TI-C** undergoes a great red shift of 140 nm compared to **TI-P** (Figure 3). In view of that the change of luminescence properties is mainly due to the differences in morphology and aggregation structure, DSC tests were conducted on different samples of **PI**, **NI**, **TI,** and **FI** to understand the differences of their varied forms. As can be seen from Figure 5, for **PI**, **NI**, **TI,** and **FI,** the melting temperatures and the integral area of the melting peak of their crystallized samples are higher/larger than that of precipitated samples. This confirms experimentally the influence of different molecular packing tightness on the luminescent properties [35].

Furthermore, single crystal analysis was carried out to figure out the effect of inter-molecular interaction on MC effect. As shown in Figure 6, **PI-C**, **NI-C**, **TI-C,** and **FI-C** all adopt the internal twisted cross molecular conformation. For **PI-C**, the dihedral angle between the adjacent rings A_1_ and B_1_ is 37.73°, allowing each benzene rings in molecule to rotate freely only to a certain extent in the solution, so the molecules could display a certain fluorescence in the appropriate solvent. When the molecules are in the aggregation state, the rings in the molecules intersect with each other at a certain angle, thus showing the AIE effect. In addition, except for the aromatic donor-acceptor interactions between indole ring, there is almost no additional aromatic donor-acceptor interactions between PI-C molecules (Figure 7a). Such a cross-accumulation mode could be classified as X-aggregation, which is an important reason for **PI-C** molecule showing AIE performance. Similarly, **NI-C** molecule shows a dihedral angle of 28.84° between the ring A_2_ and ring B_2_ (Figure 6), and no intermolecular aromatic donor-acceptor interactions exists due to its distorted structure at a minimum distance of 4.26 Å (Figure 8b). **PI-C** and **NI-C** molecules all possess both the solution luminescence and AIE properties.

In the case of **TI-C**, the dihedral angle between ring A_3_ and ring B_3_ is 34.70° (Figure 6), and except partial antiparallel aromatic donor-acceptor interactions between indole rings separated at a distance of 8.99 Å, little aromatic donor-acceptor interactions is observed due to the twisted arrangement (Figure 7c). Such a staggered parallel accumulation mode could be classified as a J-aggregate, which is different from that of **PI-C** and **NI-C**, and might be caused by the large space resistance of **TI-C** molecules and the farther distance of the intermolecular surfaces. The stacking structures of **FI-C**, as shown in Figure 7d, is similar with that of **TI-C** except its greater distance of 15.0 Å between indole ring surfaces because of the intermolecular aromatic donor-acceptor interactions [36].

According to the above analysis, the solid luminescence properties of **PI**, **NI**, **TI,** and **FI** are closely related to the J/X aggregation modes caused by the distorted structure. The precipitated samples, **PI-P**, **NI-P**, **TI-P** and **FI-P**, were made in a rapid evaporation of solvents. Thus the formation of unstable conformation with minimal hindrance is controlled dynamically, and the molecular accumulation is relatively disorderly. In such case, the solid luminescence properties depend almost entirely on the structure of the molecule itself. While for the crystalline samples of **PI-C**, **NI-C**, **TI-C,** and **FI-C**, they all show a more compact molecular stacking with thermodynamically stable conformation after the slow volatilization of solvents. Here, their luminescent properties are determined by both intermolecular interactions and intramolecular structures. In the cases of **PI**, **NI,** and **FI**, the dihedral angle between the indole units and the substituted benzene ring planes in the molecule is decreased in order to achieve the minimum distance between molecules, leading to a reduction in electron push-pull ability (Figure 8a) and thus a blue-shift of the fluorescence peaks (Figure 3). By comparison, the spatial distortion of triphenylamine group in **TI** molecules decreases when the distance between molecules is reduced, resulting in an increase of electron donation capability of triphenylamine group and electron push-pull ability of the molecule (Figure 8b), and thus a red-shift on fluorescence emission curve (Figure 3).

### 2.4. Electronic Structure

The density functional theory (DFT) computation technique was used to optimize the geometries of **PI**, **NI**, **TI,** and **FI**, and the calculated molecular orbital amplitude plots and energy levels of the HOMOs/LUMOs of **PI, NI**, **TI,** and **FI** are shown in Figure 9. It can be observed that, in the case of **PI**, **NI,** and **TI**, their LUMOs are mostly concentrated upon the indole ring with the electron-withdrawing ability (A), while HOMOs are focused separately on the three different electron-pushing moieties (D) such as 4-ethoxyphenyl, 4-(*N*,*N*-dimethylamine) phenyl, and 4-triphenylamine, verifying their status as typical D-π-A molecules. For **FI**, due to the strong electron absorption of trifluoromethyl, the polarity of the whole molecule reverses. Here, LUMO of the molecule mostly concentrates on 4-trifluoromethyl phenyl (A) while HOMO on indole mother ring (D), which displays relatively strong electron-supplying ability. The D-π-A structural feature for these four compounds corroborates the previous discussion that the positive solvatochromism effect mainly come from intramolecular charge transfer. The relatively large HOMO-LUMO gaps, 4.26 eV, 6.21 eV, 5.95 eV, and 6.82 eV corresponding respectively to **PI**, **NI**, **TI,** and **FI**, mean that these compounds could be modified by functional groups to adjust the molecular energy gaps, so as to further obtain the novel compounds with better performance.

### 2.5. pH-Sensing

Since the nitrogen atoms of the indole units acted as alkaline centers could be protonated, the effect of protonation on the optical properties of **PI**, **NI**, **TI,** and **FI** is also studied. With the addition of TFA, different samples display varied UV spectra and solution colors (Figure 10). For instance, the absorption peak of **PI** at 428 nm shifts gradually to 488 nm, and the solution color changes from yellow to purple and finally to blue (a); the absorption peak of NI appears at 460 nm and then disappears little by little, and the solution color changes from colorless to yellow and finally to colorless (b). For TI, its absorption peak at 409 nm become stronger and the solution color changes from colorless to yellow (c). While in the case of FI, there has been no absorption peak in the visible absorption region and the solution color never changes (d). The above results indicate that the four kinds of molecules have different protonation processes due to the different functional group designs, which can be clearly observed in its ultraviolet absorption spectrum and color, so this type of functional molecules can be used as a good proton display agent.

Figure 11 shows the changes of **PI**, **NI**, **TI,** and **FI** solutions in PL spectra and fluorescent colors with the addition of TFA. For **PI** (Figure 11a), the intensity of fluorescence emission peak (538 nm) initially decreases slightly. Next, the emission peak first blue shifts to 513 nm and then conversely red shifts to 525 nm with TFA concentration increasing from 5 × 10^−4^ to 5 × 10^−3^ mol/L, accompanying by a change of fluorescent colors of solution from yellow to olivine and to pink. Based on Figure 11a and the molecular structure of **PI**, it is speculated that the protonation process of **PI** is a two-stage reaction.

Similar to **PI**, **NI** also has dual response to H^+^ but after a certain degree of protonation its fluorescence intensity become too weak to show its binding to protons by fluorescence (Figure 11b). This may relate to the weaker electron supply capacity of (*N*,*N*-dimethylamine) phenyl in **NI**. **TI** and **FI** molecules show more sensitive to H^+^, and the fluorescence quenching occurs at extremely low H^+^ concentration (Figure 11c,d). For **TI**, such phenomenon may be ascribed to the destroyed conjugate structure of indole and the still rotated benzene ring in the structure of triphenylamine after binding of **TI** molecule and H^+^. While for **FI**, as the protonation occurs, the indole ring is broken, and the lower electron cloud density of the molecule itself is further decreased, thus the fluorescence disappears.

### 2.6. Fluorescent Paper

The optical property of **PI**, **NI**, **TI,** and **FI** in the solid state is also sensitive to the surrounding TFA vapor. The pieces of filter paper which have been soaked in DCM solution of **PI**, **NI**, **TI,** and **FI** (5 × 10^−3^ mol/L), called **PIP**, **NIP**, **TIP,** and **FIP**, show great changes in color and fluorescence when being exposed to TFA vapor for 5 s and 30 min, respectively (Figure 12). The color of **PIP** is almost unchanged under daylight while the fluorescence is changed from light-green to green, further to yellow-green under 365 nm ultraviolet light. For other samples, under day light, the color turns from yellow to orange for **NIP**, from white to yellow for **TIP**, and from white to pink for **FIP**. Further, under UV light, the fluorescence turned from red to golden for **NIP**, from yellow to golden for **TIP**, and from green to yellow-green for **FIP**.

The discoloration behaviors of four kinds of fluorescent papers accords with the above situations of the combination between the four molecules and proton. The result indicates that the solid samples of **PI**, **NI**, **TI,** and **FI** can respond quickly to the TFA atmosphere, and have dual color change effect under daylight and ultraviolet lamp, providing a promising potential for information encryption.

### 2.7. Logic Gate Application

Inspiring by the pH-responsive fluorescence of as-prepared indole derivatives, we further explore their possible application in logic gates. In view of the novel characteristics of **PI** solution that its emission peak positions have one-to-one corresponding relations with adding the alkali, weak acid, and strong acid, respectively, we construct a logic system. For most reported molecular logic gates, they usually are the single logic operation due to only a single signal as an output. In this work, on the basis of the pH-mediated multi-signal response, a three-input/three-output logic system is designed by rationally defining logic states, which could increase the efficiency of signal transmission.

In these logic operations, the **PI** aqueous solution serves as a gate, while alkali, weak acid, and strong acid are used as inputs and fluorescent signals as outputs. The presence and absence of inputs are defined as 1 and 0, respectively. When alkali, acid and strong acid were used as chemical inputs, the **INH** logic gate (fluorescence emission peak 513 nm as output), **INHH** logic gate (fluorescence emission peak 525 nm as output), and **IMPLICATION** (IMP) logic gate (fluorescence emission peak 538 nm as output) are constructed (Figure 13) and could display quick and sensitive optical behavior. A logic gate with three output signals would improve its stability greatly, and so far, three-output logic gates have been barely reported. Therefore, the logic system based on the pH-responsive **PI** with high sensitivity and fast response could also hold potential applications in the field of materials science and multiple signal transmission.

## 3. Materials and Methods

### 3.1. Materials and Measurements

All chemicals were purchased from Acros or Aldrich and used as received. ^1^H and ^13^C NMR spectra were collected on a Bruker Spectrospin Avance 400 MHz NMR spectrometer in CDCl_3_ with tetramethylsilane (TMS; δ = 0 ppm) as internal standard. ESI mass spectra were measured on an HPLC Q-Exactive HR-MS spectrometer (Thermo, MA, USA) by using MeOH as mobile phase. HRMS was carried out on a Micromass-GTC spectrometer. UV absorption spectra were obtained on a Varian CARY 100 Bio spectrophotometer. PL spectra were recorded on a spectrofluorophotometer (RF-5301PC, SHIMADZU, Kyoto, Japan). Differential scanning calorimetry (DSC) was conducted by PerkinElmer Pyris 1 DSC with heating rate of 10 °C/min from 20 °C to 200 °C under nitrogen atmosphere. The pictures of structure were produced using Diamond 3.1. CCDC 1842007-1842010 contained the supplementary crystallographic data of this work. The ground-state geometries were optimized with the BLYP/6-31G (d, p). Theoretical B3LYP/6-31G (d, p) density functional theory has been employed to examine the electronic properties of donor-bridge-acceptor molecular system and determine the energies [37,38,39,40,41]. The electronic states of the system have been calculated depends on Koopman’s theorem under the orbital-vertical theory.

### 3.2. Synthetic Procedures

**5-bromo-2,3,3-trimethyl-3H-indole.** The bromphenylhydrazine hydrochloride (15.0 g, 67.1 mmol) and 60 mL ethanol were added into the 250 mL single-neck bottle, to which the concentrated sulfuric acid (7.0 mL) was then slowly added in the stirring condition. The mixture was heated at 90 °C for 5 h, and allowed to cool to room temperature. After adjusting the pH of the mixture to 8.0 with Na_2_CO_3_ solution (10%), further heating was conducted to remove the solvent. The obtained residues were purified by silica gel column chromatography (eluent: PE/EA = 3), gaining red oily liquid (m = 15.50 g, yield= 97%) [42].

^1^H-NMR (400 M, CDCl_3_) δ (ppm): 7.44–7.40 (m, 1H), 7.39–7.35 (m, 2H,), 2.25 (s,3H), 1.28 (s, 6H).

^13^C-NMR (400 MHz, CDCl_3_) δ (ppm): 188.7, 152.8, 148.0, 130.8, 125.0, 121.4, 119.0, 54.3, 23.1, 15.6.

**PI.** 5-bromo-2,3,3-trimethyl-3H-indole (1.20 g, 5.04 mmol), Na_2_CO_3_ (3.48 g, 25.2 mmol), 4-ethoxyphenylboronic acid (**1a**, 2.51 g, 15.12 mmol), and Pb(PPh_3_)_4_ (0.06 g) were added into a 250 mL single-mouth bottle under the condition of nitrogen protection, to which 60 mL degassed binary solvent (40 mL THF, 20 mL H_2_O) was then introduced via syringe. Afterward, the mixture was heated at 90 °C for 12 h, and allowed to cool to room temperature. Further heating was conducted to remove the solvent, and the residues were purified by silica gel column chromatography (eluent: PE/EA = 3), gaining orange solid powder (m = 1.28 g, yield = 91%).

^1^H NMR (400 MHz, CDCl_3_) δ (ppm): 7.58 (d, J = 8.0 Hz, 1H), 7.55–7.52 (m, 1H), 7.51 (d, J = 2.1 Hz, 1H), 7.49 (dd, J = 8.0, 1.8 Hz, 1H), 7.45 (d, J = 1.4 Hz, 1H), 7.02–6.92 (m, 2H), 4.08 (q, J = 7.0 Hz, 2H), 2.32 (s, 3H), 1.44 (t, J = 7.0 Hz, 3H), 1.35 (s, 6H).

^13^C NMR (400 MHz, CDCl_3_) δ (ppm): 187.95, 158.38, 152.66, 146.27, 138.16, 133.83, 128.20, 126.25, 119.93, 119.86, 114.77, 63.53, 53.72, 23.23, 15.52, 14.90.

HRMS: Calculated = 280.1701, found = 280.1744 (M + H)^+^.

FT-IR (v/cm^−1^): 2969, 2922, 2865, 1605, 1569, 1512, 1459, 1397, 1240, 1172, 1047, 832, 801.

**NI.** Replacing **1a** by (4-(dimethylamino)phenyl)boronic acid (**2a**, 2.49g, 15.11mmol), compound **NI** was synthesized using a similar procedure described for compound **PI**. Further heating was conducted to remove the solvent, and the residues were purified by silica gel column chromatography (eluent: PE/EA = 10), gaining orange solid powder (m = 0.83 g, yield = 60%).

^1^H NMR (400 MHz, CDCl_3_) δ (ppm): 7.62 (d, J = 8.0 Hz, 1H), 7.51 (ddd, J = 16.7, 11.0, 3.4 Hz, 4H), 6.91 (s, 2H), 3.03 (s, 6H), 2.40 (s, 3H), 1.38 (s, 6H).

^13^C NMR (400 MHz, CDCl_3_) δ (ppm): 187.67, 151.89, 149.76, 146.12, 138.64, 129.69, 127.82, 125.82, 119.83, 119.49, 112.96, 53.66, 40.72, 23.25, 15.45.

HRMS: Calculated = 279.1861, found = 279.1852 (M + H)^+^.

FT-IR (v/cm^−1^): 2964, 2922, 2358, 1590, 1486, 1324, 1276, 817.

**TI.** Replacing **1a** by (4-(diphenylamino)phenyl)boronic acid (**3a**, 4.35 g, 15.05 mmol), compound **TI** was synthesized using similar procedure described for compound **PI**. Further heating was conducted to remove the solvent, and the residues were purified by silica gel column chromatography(eluent: PE/EA = 5), gaining brownish solid powder (m = 1.51 g, yield = 75%).

^1^H NMR (400 MHz, CDCl_3_) δ (ppm): 7.71 (d, J = 7.7 Hz, 1H), 7.56 (d, J = 8.1 Hz, 1H), 7.52 (s, 1H), 7.47 (d, J = 8.7 Hz, 2H), 7.30–7.26 (m, 4H), 7.19–7.09 (m, 6H), 7.05 (t, J = 7.3 Hz, 2H), 2.51 (s, 3H), 1.42 (s, 6H).

^13^C NMR (400 MHz, CDCl_3_) δ (ppm): 188.18, 152.56, 147.70, 147.03, 146.23, 138.05, 135.40, 129.20, 127.87, 126.28, 124.36, 124.04, 122.01, 119.94, 119.84, 53.74, 23.23, 15.50.

HRMS: Calculated = 403.2174, found = 403.2154 (M + H)^+^.

FT-IR (v/cm^−1^): 2959, 2917, 2366, 2343, 1591, 1491, 1334, 1272, 807, 754, 693, 608.

**FI.** Replacing **1a** by (4-(trifluoromethyl)phenyl)boronic acid (**4a**, 2.87 g, 15.10 mmol), compound **FI** was synthesized using similar procedure described for compound **PI**. Further heating was conducted to remove the solvent, and the residues were purified by silica gel column chromatography(eluent: PE/EA = 3), gaining pale yellow solid powder (m = 1.06 g, yield = 70%).

^1^H NMR (400 MHz, CDCl_3_) δ (ppm): 7.70 (s, 4H), 7.67 (d, J = 8.0 Hz, 1H), 7.57 (dd, J = 8.0, 1.8 Hz, 1H), 7.51 (d, J = 1.6 Hz, 1H), 2.39 (s, 3H), 1.39 (s, 6H).

^13^C NMR (400 MHz, CDCl_3_) δ 189.31, 153.29, 146.40, 144.87, 137.15, 129.00, 125.73, 125.69, 125.65, 122.97, 120.40, 53.89, 23.15, 15.53.

HRMS: Calculated = 304.1313, found = 304.1322 (M + H)^+^.

FT-IR (v/cm^−1^): 2970, 2868, 2362, 2336, 1610, 1569, 1463, 1320, 1156, 1120, 1064, 824.

(The detail data were shown in Appendix A).

**PI-C**: 50 mg **PI** sample was dissolved in 20 mL tetrahydrofuran/water (1:1) mixed solution in a 100mL beaker. As the solvent volatilized slowly at room temperature, the yellowish rod-like crystals were obtained.

**PI-P**: 0.10 g **PI** sample was dissolved in 10mL tetrahydrofuran solution in a 25mL round-bottomed flask, which was dried in vacuum by heating at 40 °C for 24 h, and the pale yellow solid powders was obtained.

Three other samples (**NI**, **TI** and **FI**) were used instead of **PI** to perform the above two processes, and the corresponding crystal and solid samples were obtained, respectively. Among them, **NI-C** and **NI-P** were yellow rod-like crystals and light yellow solid powders, respectively; **TI-C** and **TI-P** were red rod-like crystals and saffron yellow solid powders, respectively; **FI-C** and **FI-P** were yellow acicular crystals and beige solid powders, respectively.

## 4. Conclusions

Four novel indole derivatives, **PI**, **NI**, **TI,** and **FI**, were designed and characterized by ^1^H NMR, ^13^C NMR, HRMS, and FI-IR. The results showed that the as-prepared indole functional molecules have more diverse fluorescence properties, and their multi-level response functions can be achieved by regulating the structure of substituents. As D-π-A compounds, the solutions of **PI**, **NI**, **TI,** and **FI** show a variable red-shifted emission owing to the different push-pull electronic effects of substituents moieties. Owing to the different phenyl-substituent units as a propeller-like rotor, **PI**, **NI**, **TI,** and **FI** experience little aromatic donor-acceptor interactions in their condensed phase and thus exhibit obvious fluorescence emission in the solid-state. Additionally, their crystallization and precipitation samples show multiple chromism effects and varied fluorescence. In addition, **PI**, **NI**, **TI,** and **FI** could be easily protonated at the site of the nitrogen atom and oxygen atom, causing dramatic color and fluorescence changes, which opened up the potential avenues of developing novel colorimetric pH sensors, fluorescence paper, and logic gate applications.

## Data Availability

Appendix A: Available: Crystallographic data, ^1^H and ^13^C NMR spectra, HRMS spectra and FT-IR spectra of **PI**, **NI**, **TI,** and **FI**.

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
