# Peer review of "A Novel Indole Derivative with Superior Photophysical Performance for Fluorescent Probe, pH-Sensing, and Logic Gates"

_ijms, 2023, doi:10.3390/ijms24021711_

Round 1

Reviewer 1 Report

The present manuscript is dedicated to synthesis of several 2,3,3-trimethyl-3H-indole derivatives and studies of applications of these derivatives in different fields. Overall, scientific component of the manuscript is good, but lots of typos are presented. So, authors should carefully recheck the text. The manuscript should be corrected according to the following notes:

1) Term “indolium” corresponds to charged indole. In the manuscript all presented derivatives contain 2,3,3-trimethyl-3H-indole unit but not indolium. Authors should correct title and text.

2) What method of ionization was used for HRMS, it should be pointed in Experimental

3) For 5-bromo-2,3,3-trimethyl-3H-indole any physical constants or 1H-NMR should be given.

4) For compound N1 an extra 13C signal is presented. For compound T1 2 13C signals are missed. For compound F1 19F NMR should be provided. In 13C spectra 6 extra signals are presented. Moreover, 13C signal of CF3 is triplet with high C-F coupling constant. Authors should carefully check spectral data. Unfortunately, no supporting information was attached for review. I hope, it would be available on second round of revision.

5) What about fluorescence quantum yields in solution for the compounds? It should be measured and discussed.

Author Response

Reviewer 1

Comments:

The present manuscript is dedicated to synthesis of several 2,3,3-trimethyl-3H-indole derivatives and studies of applications of these derivatives in different fields. Overall, scientific component of the manuscript is good, but lots of typos are presented. So, authors should carefully recheck the text. The manuscript should be corrected according to the following notes:

We thank the reviewer for the valuable comments and suggestions.

  1. Term “indolium” corresponds to charged indole. In the manuscript all presented derivatives contain 2,3,3-trimethyl-3H-indole unit but not indolium. Authors should correct title and text.

Thanks. We now change the all "indolium" to "indole" in manuscript and SI.

  1. What method of ionization was used for HRMS, it should be pointed in Experimental.

Thank you very much for your comments. ESI mass spectra were measured on a HPLC Q-Exactive HR-MS spectrometer (Thermo, USA) by using MeOH as a mobile phase. According to the reviewer’s suggestion, we add in the Experimental section. Please see the revised manuscript.

  1. For 5-bromo-2,3,3-trimethyl-3H-indole any physical constants or 1H-NMR should be given.

We now add the the 1H-NMR and 13C NMR of 5-bromo-2,3,3-trimethyl-3H-indole in the revised manuscript.

1H-NMR (400 M, CDCl3) δ (ppm): 7.44-7.40 (m, 1H), 7.39-7.35 (m, 2H,), 2.25 (s,3H), 1.28 (s, 6H).

13C-NMR (400 MHz, CDCl3) δ 188.7, 152.8, 148.0, 130.8, 125.0, 121.4, 119.0, 54.3, 23.1, 15.6.

  1. For compound N1 an extra 13C signal is presented. For compound T1 2 13C signals are missed. For compound F1 19F NMR should be provided. In 13C spectra 6 extra signals are presented. Moreover, 13C signal of CF3 is triplet with high C-F coupling constant. Authors should carefully check spectral data. Unfortunately, no supporting information was attached for review. I hope, it would be available on second round of revision.

We thank the reviewer for the valuable comments and suggestions. We now carefully checked 13C NMR of NI, added the 13C NMR of TI and spectra 6, and revised the spectral data. Please see the revised manuscript. We are sorry that the reviewer cannot find the SI in first verion, now we conbine the main text and SI in the revised version.

For compound N1 an extra 13C signal is presented.

we carefully checked the NI. 13C NMR (400 MHz, CDCl3) δ (ppm): 187.67, 151.89, 149.76, 146.12, 138.64, 129.69, 127.82, 125.82, 119.83, 119.49, 112.96, 53.66, 40.72, 23.25, 15.45.

For compound T1 2 13C signals are missed.

We checked the TI. 13C NMR (400 MHz, CDCl3) δ (ppm): 188.18, 152.56, 147.70, 147.03, 146.23, 138.05, 135.40, 129.20, 127.87, 126.28, 124.36, 124.04, 122.01, 119.94, 119.84, 53.74, 23.23, 15.50.

For compound F1 19F NMR should be provided.

All the intermediates and final products were purified carefully and characterized fully by 1H NMR, 13C NMR, HRMS and IR. We think that the structure of FI was confirmed.

In 13C spectra 6 extra signals are presented.

13C NMR (400 MHz, CDCl3) δ (ppm): 189.31, 153.29, 146.40, 144.87, 137.15, 129.00, 125.73, 125.69, 125.65, 122.97, 120.40, 53.89, 23.15, 15.53.

  1. What about fluorescence quantum yields in solution for the compounds? It should be measured and discussed.

Thanks for the reviewer’s suggestion. In current work, we observed the fluorescence quantum properites, but we does not study the fluorescence quantum yield in solution in depth. It could be a very interesting topic in our future work. Thanks again for the reviewer valuable comments.

Reviewer 2 Report

This work make no sence to the readers. And these probes could do noting, and no cell experiments was performed. I suggest rejection of this paper.

Author Response

Reviewer 2

Comments:

This work make no sence to the readers. And these probes could do noting, and no cell experiments was performed. I suggest rejection of this paper.

We thank the reviewer for the critical comments. However, the reviewer does not carefully review this paper and have a prejudice. In this study, we synthesized the four novel indoliumindoleindole derivatives on the basis of the donor-π-acceptor (D-π-A) concept. These derivatives exhibit positive solvatochromism owing to their varied molecular conformations upon contacting to various solvents and the different HOMO-LUMO gaps caused by the difference in electronic push-pull capability of the substituents. Their solid-state fluorescence emissions and multiple chromisms are observed due to the inherent twisted geometries and aggregation modes. In addition, these derivatives show dramatic color and fluorescence responses and can be a potential novel colorimetric pH sensors, fluorescent papers and logic gates.

We believe this novel and multidisciplinary topic will be of very interest to a broad scientific readership working in the fields of materials science, medicine, physics, nanoscience, and chemistry.

In current state, the probe can be used as the chemical sensor. We think it is not necessary to perform the cell experiment. However, we still thank the reviewer for the valuable comments, the cell experiment is necessary as a biosensor in our future work.

Reviewer 3 Report

In paper “A Novel Indolium Derivative with Superior Photophysical Performance for Fluorescent Probe, pH-Sensing, and Logic gates” by Hai-Ling Liu, four novel indolium derivatives, PI, NI, TI and FI, were designed and characterized by various experimental techniques, and the conformational and electronic properties were calculated by using DFT.

The experimental results show that the as-prepared molecules have more diverse fluorescence properties regulated by the substituents. The solutions of these D-π-A mlecules show a variable red-shifted emission. The conformation of biphehyl unit prevents efficient π-π stacking interaction in the condensed phases, leading to fluorescence emission in the solid-state. Crystallized and precipitated samples have been studied as well for their multiple chromism effects and varied fluorescence. Possible applications of newly synthesized samples as colorimetric pH sensors, fluorescence paper and logic gates have been demonstrated.

In its present form I cannot recommend this manuscript for publication. The following questions should be answered and the manuscript should be revised accordingly.

·        Please check this sentence, the meaning of which is vanishing: Only a few reports that used them as multilevel signal probes in the complex detection environments, which could provide accurate analysis.

·        Scheme 1. I do not understand the appearance of this scheme here, it looks like graphical abstract. If not, please provide description of each part, and also explain why parts of this scheme are appearing as separated figures later.

·        B3LYP/6-31G (d) is not a program, but functional and basis set. Please justify their usage for all four molecules

·        the authors have to explain the citation of those references. At first glance they appear here by a mistake, since they are not proper citation to Gaussian, functional or basis set. The authors have to replace them by proper citations.

·        Line 183 “the four compounds possess different degrees of solvation effect” – which degrees are meant here?

·        The next line: first ,the authors should explain why to use a poor solvent, and then please explain “excited vibration states with similar energy due to the poor 186 solubility of molecules in hexane” for the particular case of the studied molecules, without a reference, where not much is said about excited vibration states with similar energy – please also check the reference or provide another one.

·        On line 242 the authors start explanations of the stacking patterns by saying “Furthermore, single crystal analysis was carried out to figure out the effect of intermolecular π-π stacking on MC effect…” First, not much difference is found in the dihedral. How authors can explain this? Second, It is quite strange to hear a term "pi-pi stacking distance" if this distance exceeds 3.6 A. The authors have to reconsider terminology they have used. Third, which intermolecular interactions to the opinion of the authors could be realized for the molecules FI-C at such large (15 A) intermolecular distance?

·        Figure9: Here I agree with the authors, that TI shows indeed D-pi-A structure, and it is reflected in the HOMO/LUMO localizations. At the same time, for three remaining compounds this is hardly visible. More precise description is expected in the manuscript.

·        In conclusions, the authors are invited to discuss the correlation of the properties in the row of the compounds depending on the strength of donor and acceptor groups they have added to the common core of the compounds.

·         

Author Response

Reviewer 3

Comments:

In paper “A Novel Indolium Derivative with Superior Photophysical Performance for Fluorescent Probe, pH-Sensing, and Logic gates” by Hai-Ling Liu, four novel indolium derivatives, PI, NI, TI and FI, were designed and characterized by various experimental techniques, and the conformational and electronic properties were calculated by using DFT.

The experimental results show that the as-prepared molecules have more diverse fluorescence properties regulated by the substituents. The solutions of these D-π-A mlecules show a variable red-shifted emission. The conformation of biphehyl unit prevents efficient π-π stacking interaction in the condensed phases, leading to fluorescence emission in the solid-state. Crystallized and precipitated samples have been studied as well for their multiple chromism effects and varied fluorescence. Possible applications of newly synthesized samples as colorimetric pH sensors, fluorescence paper and logic gates have been demonstrated.

In its present form I cannot recommend this manuscript for publication. The following questions should be answered and the manuscript should be revised accordingly.

Thank you very much for your kind comments.

  1. Please check this sentence, the meaning of which is vanishing: Only a few reports that used them as multilevel signal probes in the complex detection environments, which could provide accurate analysis.

Thanks, we now modified to “Indolium probes that enable accurate detection in complex environments are rarely reported”.

  1. Scheme 1. I do not understand the appearance of this scheme here, it looks like graphical abstract. If not, please provide description of each part, and also explain why parts of this scheme are appearing as separated figures later.

Thank you very much for your valuable comments, we now delete the Scheme 1 in manuscript and use as a graphic abstract.

  1. B3LYP/6-31G (d) is not a program, but functional and basis set. Please justify their usage for all four molecules. the authors have to explain the citation of those references. At first glance they appear here by a mistake, since they are not proper citation to Gaussian, functional or basis set. The authors have to replace them by proper citations.

Thanks, the expression of B3LYP/6-31G (d) is not correct, we now change to “The ground-state geometries were optimized with the B3LYP/6-31G (d, p)”. Please see the reference “Bridge efect on the charge transfer and optoelectronic properties of triphenylaminebased organic dye sensitized solar cells: theoretical approach, Malak Lazrak, Hamid Toufik, Si Mohamed Bouzzine, Fatima Lamchouri, Res. Chem. Intermed., 2020, 46, 3961-3978.” We now replace the reference.

  1. Line 183 “the four compounds possess different degrees of solvation effect” – which degrees are meant here?

The "degree" here refers to the different fluorescence properties shown in different solutions. With increasing solvent polarity, red-shifted emissions with varying degrees are observed in mid-polar solvents (EA, THF and DCM) and polar solvents (DMF and acetonitrile). For instance, the fluorescence peaks of PI, NI, TI and FI are red shifted by 50 nm, 29 nm, 55 nm and 57 nm respectively.

  1. The next line: first, the authors should explain why to use a poor solvent, and then please explain “excited vibration states with similar energy due to the poor 186 solubility of molecules in hexane” for the particular case of the studied molecules, without a reference, where not much is said about excited vibration states with similar energy – please also check the reference or provide another one.

Thanks for the reviewer’s professional comments. The use of solvents of different polarities (including hexane) can more deeply excavate the fluorescence properties of molecules affected by the environment. The choice of solvent is on the basis of a reference "Y. Lu, H. Yan, X. Meng, B. Li and S. Ge, J. Mater. Chem. C, 2017, 5, 10589-10599”.

The original description “excited vibration states with similar energy due to the poor solubility of molecules in hexane” may be not correct. We now modified to "which indicated two or three close-lying excited vibration states with small energy gap."

We here provide another reference "Y. Lu, H. Yan, X. Meng, B. Li and S. Ge, J. Mater. Chem. C, 2017, 5, 10589-10599”. Please see ref 33 in the revised manuscript.

  1. On line 242 the authors start explanations of the stacking patterns by saying “Furthermore, single crystal analysis was carried out to figure out the effect of intermolecular π-π stacking on MC effect…” First, not much difference is found in the dihedral. How authors can explain this? Second, It is quite strange to hear a term "pi-pi stacking distance" if this distance exceeds 3.6 A. The authors have to reconsider terminology they have used. Third, which intermolecular interactions to the opinion of the authors could be realized for the molecules FI-C at such large (15 A) intermolecular distance?

Thanks for reviewer’s professional and valuable comments. The biggest difference is the dihedral angle between the indole units and the substituted benzene ring planes in the molecules, which is the key reason for the difference in molecular fluorescence phenomena.

The distance between molecules is greater than 3.6A, as it turns out, these molecules exhibit the AIE phenomenon, and we propose that there is still a small pi-pi stacking effect between molecules. and we the expression of "pi-pi stacking distance" might not correct, we now modify in the revised manuscript.

We proposed that the CF3 group produces an intermolecular force similar to hydrogen bonding due to its strong electrical absorption. In this case, the dihedral angle of FI-C between the indole units and the substituted benzene ring planes in the molecule are affected, resulting in a change in fluorescence performance. However, this conjecture has not been proved experimentally, and we think that further research can be carried out in subsequent work.

  1. Figure9: Here I agree with the authors, that TI shows indeed D-pi-A structure, and it is reflected in the HOMO/LUMO localizations. At the same time, for three remaining compounds this is hardly visible. More precise description is expected in the manuscript.

Thanks for reviewer’s positive comments. We think that these four molecules are relatively small, and their D-pi-A structure may not be obvious in the current HOMO/LUMO localizations, and their characteristics may be better revealed by optimizing the parameters or methods of simulation calculations. On the other hand, their performance back-demonstrates that they have D-pi-A properties.

  1. In conclusions, the authors are invited to discuss the correlation of the properties in the row of the compounds depending on the strength of donor and acceptor groups they have added to the common core of the compounds.

Thank you very much for you professional comments. We will continue our future work on the relation between the properties and donor and acceptor functional groups. Thanks again!

Round 2

Reviewer 1 Report

Authors made the appropriate corrections.

Author Response

Thank you!

Reviewer 2 Report

The author claimed that "In this study, we synthesized the four novel indoliumindoleindole
derivatives on the basis of the donor-π-acceptor (D-π-A) concept.
" New compounds based on D-π-A are enormous, this could not be the originality. And the author said that this is fluorescent probe, but for detection what? Just pH? We have pH test paper, and the probe could detect cell samples, I see no significance for developing the probe, the author said that" the reviewer does not carefully review this paper and have a prejudice ", as a reviewer I reviewed more than 100 papers till now, and I never reviewed paper with prejudice, this paper just lack novelty and experiments demnonstrate nothing  and make no sense to readers. 

Author Response

We thank the reviewer for the critical comments. However, the reviewer does not carefully review this paper and have a prejudice. In this study, we synthesized the four novel indoliumindoleindole derivatives on the basis of the donor-π-acceptor (D-π-A) concept. These derivatives exhibit positive solvatochromism owing to their varied molecular conformations upon contacting to various solvents and the different HOMO-LUMO gaps caused by the difference in electronic push-pull capability of the substituents. Their solid-state fluorescence emissions and multiple chromisms are observed due to the inherent twisted geometries and aggregation modes. In addition, these derivatives show dramatic color and fluorescence responses and can be a potential novel colorimetric pH sensors, fluorescent papers and logic gates.

We believe this novel and multidisciplinary topic will be of very interest to a broad scientific readership working in the fields of materials science, medicine, physics, nanoscience, and chemistry.

In current state, the probe can be used as the chemical sensor. We think it is not necessary to perform the cell experiment. However, we still thank the reviewer for the valuable comments, the cell experiment is necessary as a biosensor in our future work.

Reviewer 3 Report

Dear Authors, unfortunately you preferred to skip work on the most important questions. Here, I will repeat them, and ask you to take them seriously, since you made some conclusions which are not supported by your data. 

1. B3LYP/6-31G is functional and the authors have now changed the basis set. This has to be named properly in the text. The reference, why the usage of this functional for these new compounds is undertaken should be included.

The referemce to GAUSSIAN has to appear as well. 

2. The question about hexane, obviously a poor solvent for these componds, which leads to the molecular aggregation, and as a consequence to the changes in optical properties, remains. Please provide comment, not just a reference. 

3. "a small pi-pi stacking effect between molecules", as the authors commented, cannot be realized on the distances greater that I have provided. Even for some compounds (from XRD data) shorter distances do not guarantee the pi-stacking interactions. I am asking the authors provide a clear explanation which interactions can be realized through space on distances greated the pi-stacking distances. For some compounds the authors have depicted, these distances are far beyound the size of the molecule istelf. At the same time, the authors name them "staking patterns" or pairs, a part of cristalline packings. What is then in empty spaces between the molecules? Are the authors sure about their conclsions? This point is the most critical one.

4.  ..."CF3 group produces an intermolecular force similar to hydrogen bonding" - please provide a partner for this H bond, provide also the distances at which it is realized.

5. ..."D-pi-A structure may not be obvious in the current HOMO/LUMO localizations" - Here, I disagree with the authors. The molecles have size which is OK to see the delocalizations, if the correct functional is set for the calculations. The selected B3LYP is know for its overestimation of the delocalization. Therefore, another functional and basis set has to be set for the calculations. the authors are invited to perform new computational work here.

6. "In conclusions, the authors are invited to discuss the correlation of the properties in the row of the compounds depending on the strength of donor and acceptor groups they have added to the common core of the compounds" - this comment from the previous round was ignored. I apologize if the authors have not got my appeal -  was asking to include in the conclusion of their paper already now, in this manuscript, the description of some patterns, dependencies they observed.

Round 3

Reviewer 2 Report

accept

Author Response

Thank you very much for your kind comments

Reviewer 3 Report

1. I disagree with the authors, when they say that CF3 group interacting with some other group can be classified as "aromatic donor–acceptor (D-A)", since CF3 group does not have an aromatic character. Please clarify.

2. I still disagree with "a new term" used by the auhors, claiming that aromatic donor–acceptor (D-A) can be realized through space on such long distances. I insist on another explanation of the structures. Please also refer to this link to see that aromatic donor–acceptor (D-A) intearctions are short-range ones.

https://pubs.acs.org/doi/10.1021/acs.orglett.8b03824

Author Response

Comments:

  1. I disagree with the authors, when they say that CF3 group interacting with some other group can be classified as "aromatic donoracceptor (D-A)", since CF3 group does not have an aromatic character. Please clarify.

Thank you for your comments. We applogy that we didn't explain the question clearly. We believe that in the structure of FI molecules, 4-trifluoromethyl phenyl which has aromatic character acts as an aromatic acceptor rather than just trifluoromethyl, while the indole ring acts as an aromatic donor. On the basis of these, we believe that there is an aromatic donor–acceptor (D-A) interaction in FI.

  1. I still disagree with "a new term" used by the auhors, claiming that aromatic donoracceptor (D-A) can be realized through space on such long distances. I insist on another explanation of the structures. Please also refer to this link to see that aromatic donoracceptor (D-A) intearctions are short-range ones.

https://pubs.acs.org/doi/10.1021/acs.orglett.8b03824

Thanks for your comments. we agree with your statement that "aromatic donor–acceptor (D-A) interactions are short-range ones". But we should know that, two parts are included in the FI molecule, 4-trifluoromethyl phenyl and indole ring. In this article, in order to maintain consistency throughout the text, as with the PI, NI, TI molecules, we only focus on the distance between indole rings, which is shown to be 15Å. However, unlike the PI, NI, and TI molecules, the donor-acceptor of FI has been reversed, and indole ring group are no longer aromatic acceptors, but aromatic donors.

Therefore, the distance between the 4-trifluoromethyl phenyl groups that act as aromatic acceptors, and even the distance between donor and acceptor, should be taken into account. We believe that the distance between 4-trifluoromethyl phenyl groups and the distance between indole ring and 4-trifluoromethyl phenyl should be compatible with aromatic donor–acceptor (D-A) interactions.

As this is not the focus of this article, it can be further studied in subsequent work. Aromatic donor–acceptor (D-A) interactions has a lot to do with the nature of both donor and acceptor, and the author will discuss this in more depth in subsequent work, thanks again for the suggestions.

Round 4

Reviewer 3 Report

none